# Evolutionary Emergence of Neurodynamic Networks for Robust Control: A Simple Excitatory-Inhibitory Network

## Abstract

Fine-grained network models based on differential equations, and neurodynamic synapses and neurons provide a realistic description of biological neuronal networks, compared with mainstream artificial neural networks. They nevertheless have not been widely explored, mainly due to the lack of effective parameter training methods. We propose a neurodynamic model training method that combines an efficient neurodynamic simulation architecture and an evolutionary algorithm. Based on a simple Excitatory-Inhibitory network, a neurodynamic model with task control capabilities is successfully obtained via parallel dynamic simulation, and network selection methods under evolutionary pressure. Compared with the state-of-the-art reinforcement learning methods, the resulting neurodynamic network can achieve comparable task control performance for Mojoco tasks in a significantly smaller network scale within fewer training steps. Our work provides an alternative path to functional networks alongside mainstream reinforcement learning frameworks, and prove the feasibility of the evolutionary approach toward biological intelligence.

## 1 Introduction

Living organisms in nature can rapidly adapt to the complex environment while interacting with it in the absence of explicit supervision conditions under evolutionary pressure, where the neural systems play an important role in continuous evolution and adaptation to the environment. Early efforts on modelling biological neural networks focused extensively on the dynamics of neuronal electrophysiological activities, Hodgkin & Huxley (1952) aiming to simulate biological mechanisms. However, due to the limitations on computational resources at the time, the large-scale applications of bio-inspired dynamic models were challenging. Therefore more simplified artificial neural network (ANN) was proposed, McCulloch & Pitts (1943) and has been extensively used up to date. With the introduction of backpropagation algorithms, ANNs have proved their capabilities in several application scenario. LeCun et al. (2015) ANNs rely on the training using large-scale labelled data, with the static input-output mapping as the main processing method, Zador (2019) which abandons the rich temporal dynamics in biological neural systems. Conventional ANNs have not only succeeded in supervised learning tasks, their applications in reinforcement learning have also been widely explored, especially in scenarios where intelligences need to continuously optimise their strategies while interacting with the external environment. In fact, the central idea of reinforcement learning is essentially derived from the abstraction of the adaptive behaviour of living organisms: instead of relying on large-scale labelled samples, living organisms develop effective interacting strategies through continuous interaction with the environment, and shape by natural selection. This capability allows living organisms to process certain structures *a priori*, and learning rules, and thus to rapidly develop functional models under limited resources.

The modelling on the dynamics of neural systems has increasingly drawn attention in recent years. Van and Sompolinsky proposed the excitation-inhibition balance network, Vreeswijk & Sompolinsky (1996) which possesses rich dynamical features, Brunel (2000) and explains the experimentally observed irregular network firing under constant external input. Subsequent studies employed the excitation-inhibition balance as a basic principle to construct computational models for different functional areas of the human brain, including visual cortex Potjans & Diesmann (2014); Schmidt

et al. (2018), motor cortex Hennequin et al. (2014) and decision-making cortex Wong & Wang (2006). Later, the spiking neural network (SNN) Maass (1997) and continuous-time neurodynamic models were proposed to model the biological neural activity. The effective training of neurodynamic models, however, still faces several challenges, since they are normally highly non-linear with complex gradient propagation paths, and models driven by pulse events, *e.g.*, SNNs, are inherently non-differentiable, making the direct applications of backpropagation algorithms impossible. Several studies have attempted to introduce surrogate gradient, Neftci et al. (2019) or local learning rules Markram et al. (2012) to improve the trainability, however, these approaches still have significant gaps in terms of convergence speed, stability, and task adaptability, compared with the mainstream deep learning frameworks. Therefore, the development of new training paradigms suitable for neurodynamic models remains a key path to bring artificial systems closer to biological intelligence.

Evolutionary algorithm (EA) Sampson (1976) is inspired from the biological evolution mechanism, and has been applied to neural network training and strategy optimisation. EA does not rely on backpropagation or exact gradient information, enabling neural networks to gradually evolve functionally stable networks in complex time-varying environments. Its application on the training of neural dynamical systems, however, still remains difficult. Dynamic systems are temporally highly coupled, and the state evolution process relies on continuous numerical integration, or impulse-driven updating, requiring significantly higher computational overheads than static feed-forward networks. Therefore traditional evolutionary algorithms are normally unable to meet the training requirements in terms of efficiency and scalability, and the effective applications of evolutionary algorithm optimisation in dynamic models have long been limited by computational resources.

Evolutionary optimisation of biologically inspired networks has been explored in several recent studies. Shen et al. (2023) evolved spiking microcircuits for control tasks, demonstrating the feasibility of structural evolution in SNNs. Habashy et al. (2024) analysed how heterogeneous neuronal time constants shape functional dynamics, suggesting the importance of biophysical diversity. Najarro & Risi (2020) employed evolution to meta-optimise Hebbian plasticity rules for deep networks, highlighting an alternative route toward biologically motivated learning. Our work differs from these approaches by jointly optimising synaptic delays and weights at scale, enforcing E–I balanced population structure, and integrating these components into a unified neurodynamic model evaluated on continuous-control tasks.

In the current study, we modify the evolutionary algorithm optimisation on neurodynamic systems, and propose a novel evolutionary training paradigm for neurodynamic models. The adaptation of individuals while interacting with the environment is simulated by evolving a dynamic model over time in a reinforcement learning environment, and the evolution of a population of individuals is simulated by a parallelised evolutionary algorithm, following a bio-inspired optimisation strategy. We introduce a high-performance dynamical simulation framework ENLARGE, Qu et al. (2023) which is based on a fine-grained network representation and hierarchical communication architecture, and redesign the data structure of neuron and synapse model to preserve the biological features of the network. We also compare two efficient fitness assessment strategy, the Covariance Matrix Adaptation Evolution Strategy (CMA-ES) Hansen et al. (2003) and the Natural Evolution Strategy (NES), Wierstra et al. (2014) and parallelise the network parameter optimisation in the evolutionary algorithm to further enhance its efficiency. Our model starts from an initial network with excitatory and inhibitory (E-I) neurons, and gradually evolves a functional network with control capability.

**Contributions**  The contributions of this paper can be summarised as follows:

- Different from the existing ANN and SNN models, we propose a novel biological neurodynamic network (BNN) model to describe intelligent behavior.
- Integrating efficient dynamical simulation with parallel evolutionary algorithms accelerates BNN training and improves sampling efficiency in reinforcement learning.
- The BNN achieves comparable performance across MuJoCo tasks with significantly fewer training steps and a much smaller network size.

The aim of our study is not to surpass state-of-the-art performance in any given reinforcement learning task. Instead, we present a new modelling and training framework that provides a parallel path to existing mainstream reinforcement learning frameworks from a self-organisation perspective. As

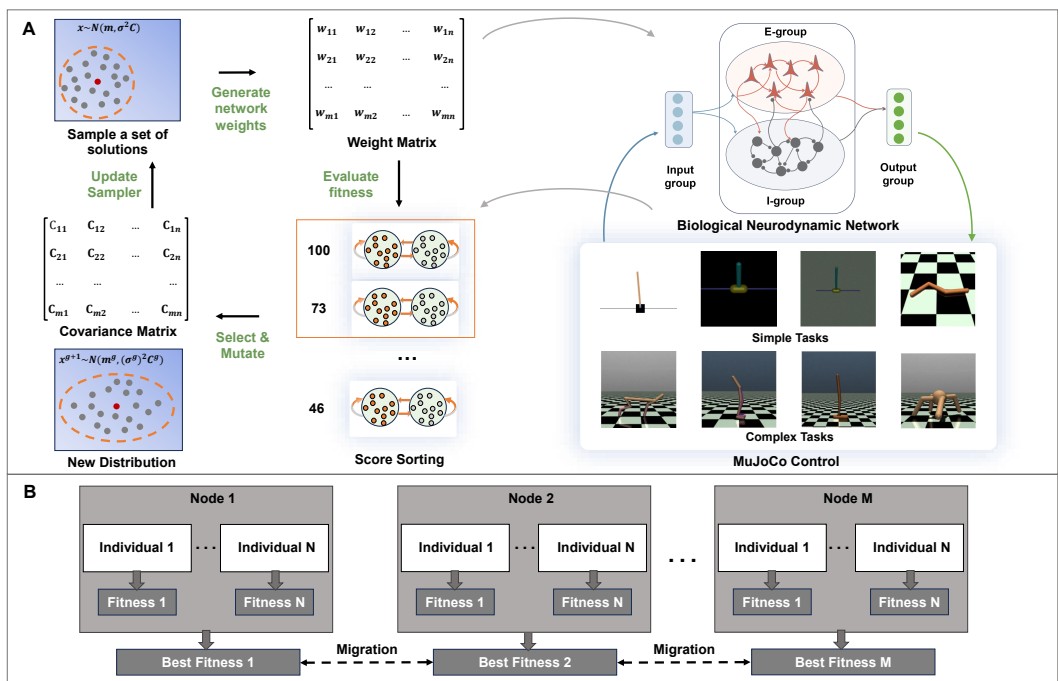

Figure 1: Illustration of the training paradigm.(A) The flowchart of the Biological Neurodynamic Network (BNN) optimisation. The model is optimised using the evolutionary algorithm starts by sampling a set of solutions to generate the parameter matrix. The fitness of the dynamics network is then evaluated through the interaction of reinforcement learning with Mujoco tasks of different difficulties. The parameters of the the distribution are updated using selection and variation operators as the next generation in the evolutionary algorithm. (B) Parallel accelerated framework. $M$ nodes are set up in the cluster, and $N$ tasks are assigned to each node according to the population size. The optimal fitness is computed for each node in each generation. The optimal parameters of the population are exchanged at a fixed interval, *i.e.*, Migration.

.

a proof-of-principle study, we successfully achieved the evolution of the neurodynamic model in a real reinforcement learning environment, thus providing an alternative path for building interpretable and evolvable brain-like intelligent systems.

## 2 RELATED WORKS

**Neurodynamic Models**    Since the pioneer works by Hodgkin and Huxley, Hodgkin & Huxley (1952) the concept of neurodynamics has been raised and studied extensively. Most studies employ a set of differential equations that describes the propagation of empirical action potential data, and the model size is normally limited to few neurons. Most neurodynamic models focused on the dynamic features, and little has been attempted to account for the behaviour. Artificial neural networks were proposed with similar origins. McCulloch & Pitts (1943) However, they quickly focused on describing the behaviour; artificial neural networks have thus diverged from neurodynamic models. Newell et al. (1958); Rosenblatt (1958) The limited understanding of the neural basis of behaviour at the time left artificial intelligence and machine learning largely disconnected from neuroscience. With the development of neuroscience and the modern computation industry, attempts have been made to propose more biologically realistic models, where a central focus is the dynamic features, which have mostly been ignored by artificial neural networks. A number of neurodynamic models Eliasmith & Anderson (2003) have been proposed, which provide an alternative path to describe intelligent behaviour.

Table 1: A comparison of the performances of different optimisation algorithms on Mujoco tasks reported in the literature. Schulman et al. (2017); Salimans et al. (2017b)

| Optimisation Approach | Gradient-based (PPO) | Evolution Strategies |
|---|---|---|
| Environment Step(s) | $10^6$ | $10^7$ |
| Benchmark Model/Size | MLP/(64,64) | MLP/(64,64) |
| **MuJoCo Task** | **Rewards** | |
| HalfCheetah | $\sim 2000$ | $\sim 4500$ |
| Hopper | $\sim 2200$ | $\sim 2100$ |
| InvertedDoublePendulum | $\sim 8000$ | $\sim 7000$ |
| InvertedPendulum | 1000 | 1000 |
| Swimmer | $\sim 120$ | $\sim 121$ |
| Walker2d | $\sim 2500$ | $\sim 2500$ |

**Gradient-based optimisation methods** have been widely used in deep reinforcement learning (DRL) in recent years, driving rapid development from gaming intelligences, *e.g.*, Atari, Go, to robotic control systems. Deep Q-Network (DQN) Mnih et al. (2015) was the first to combine convolutional neural networks and Q-learning to achieve end-to-end effective learning strategies from high-dimensional pixel inputs. Subsequently, algorithms such as Trust Region Policy Optimization (TRPO), Schulman et al. (2015) and Proximal Policy Optimization (PPO) Schulman et al. (2017) have improved deep learning in terms of convergence efficiency and policy stability, and become the current mainstream policy optimisation method. However, the success of such frameworks relies heavily on highly engineered network structures and training techniques, with limited interpretability. When confronted with more biologically realistic neural systems, these methods face difficulties such as instability, difficult or unfeasible gradient computation, and the likelihood of being trapped at local optima, thus severely limiting the trainability and convergence of the model.

**Evolutionary algorithm** (EA), as a population-based, gradient-free evolutionary optimisation strategy, it provides a robust and scalable alternative for network parameter optimisation. It has been shown that employing evolutionary algorithms to train policy networks can achieve performance comparable to mainstream gradient-based algorithms in reinforcement learning tasks. Such et al. (2018) Furthermore, structural evolution methods such as HyperNEAT Stanley et al. (2009) demonstrate the advantages of evolutionary algorithms in terms of structural interpretability and modularity generation. Stanley & Miikkulainen (2002) EAs have unique advantages in the optimisation of neurodynamic models as they do not rely on gradient information. Few works have attempted to apply evolutionary algorithms to neurodynamic network training, Mozafari et al. (2019) they nevertheless are still unable to present a systematic solution. A comparison of the performance of optimisation methods is presented in table 1. Both optimisation algorithms can achieve excellent performances on MuJoCo tasks, but they require a complex network consisting of hundreds of neurons, and millions of optimisation steps. For comparison, the main model described in this paper contains only 20 neurons, and can achieve comparable results using an evolutionary algorithm trained for $5 \times 10^5$ steps.

## 3 PRELIMINARIES

**E-I balance**   Neurons and synapses in the brain coordinate their excitatory and inhibitory inputs to establish and maintain a constant excitation-inhibition (E-I) ratio, which is known as E-I balance. Shadlen & Newsome (1994) The excitation-inhibition balance is one of the fundamental properties of the cerebral cortex, and it has been shown that this balance is prevalent in a wide range of cortical areas. Isaacson & Scanziani (2011); Haider & McCormick (2009) E-I balance also benefits the precision and efficiency of neuronal coding mechanisms, Denève & Machens (2016) extensive dynamic simulations further reveal that the E-I balanced structure is capable of generating rich computational dynamic behaviours, Vreeswijk & Sompolinsky (1996); Brunel (2000); Denève et al. (2017) and thus acts as potential building blocks of brain functions. Furthermore, recent studies suggest that training an E-I balanced network as the initial state can significantly improve the learning efficiency and stability of the network. Song et al. (2016); Ingrosso & Abbott (2019)

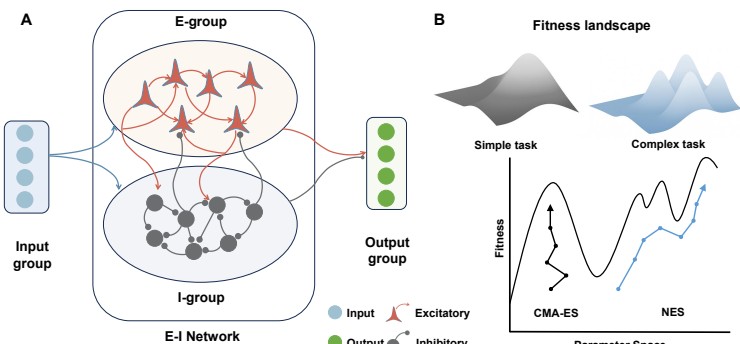

Figure 2: Illustration of the BNN architecture, and Evolutionary Algorithms. (A) Neurons are grouped into four types: input, excitatory (E), inhibitory (I), and output, coloured in blue, red, gray and green, respectively. Connections are established between neurons in all groups via two types of synapses: excitatory and inhibitory, coloured in red and gray, respectively. (B) Illustration of two evolutionary algorithms. CMA-ES is more efficient on optimisation on simple energy landscapes, whereas NES is fundamentally more suitable for multifunnel landscapes.

Beyond biological plausibility, balanced E–I networks also offer concrete functional advantages in machine learning contexts. Prior theoretical and computational studies (Wang (1999); Brunel (2000)) demonstrate that E–I balance stabilises network dynamics, speeds convergence toward attractor states, and increases robustness to noisy inputs. More recent work shows that initializing networks in an E–I balanced regime can substantially reduce the number of training steps needed to reach competent performance by constraining neural activity within dynamically stable regions of state space. These considerations motivate the inclusion of E–I balance in our evolutionary framework.

**Spiking Neural Networks**   As the most widely used model in computational neuroscience and neuromorphic computing, the Spiking Neural Network (SNN) Maass (1997); Maass & Markram (2004) provides a realistic whilst computationally feasible description of biological neural networks, compared with traditional neural networks. In a SNN, neurons are connected with each other arbitrarily via synapses, and the states of neurons and synapses are updated in cycles by firing a large number of spikes as signals. The biologically relevant features of SNN, Winer (1993) however, make the direct application of SNNs in high performance computation difficult. Many computational frameworks have been proposed to address the efficiency problem, *e.g.*, NEST, Gewaltig & Diesmann (2007) GeNN, Yavuz et al. (2016) and ENLARGE. Qu et al. (2023) In the current work, we modify the traditional SNN in the ENLARGE framework, such that continuous postsynaptic currents instead of discrete spikes are transmitted across the networks of neurons and synapses.

## 4 METHODS

### 4.1 BIOLOGICAL NEURODYNAMIC NETWORK

Our biological neuraldynamic network (BNN) is based on an Excitatory-Inhibitory network, Vreeswijk & Sompolinsky (1996) consisting of the Leaky Integrate-and-Fire (LIF) neuron model and AMPA/GABA synapses, Gerstner et al. (2014) with some modifications for better biological interoperability and viability.

**Model for neuron**   Neurodynamic models require neuron models to describe the dynamics of neurons, *i.e.*, signal firing. Multiple models for neurons have been proposed; among them, the Leaky Integrate-and-Fire (LIF) neuron has been proved to be a biologically realistic and computationally feasible one. The majority of SNNs are established by deterministic neurons, whereas biological neurons have inherent randomness in terms of firing. Ma et al. (2023) We introduce a novel **noisy input current term** into LIF neurons as an analogue to the random disturbances of real biological neurons.

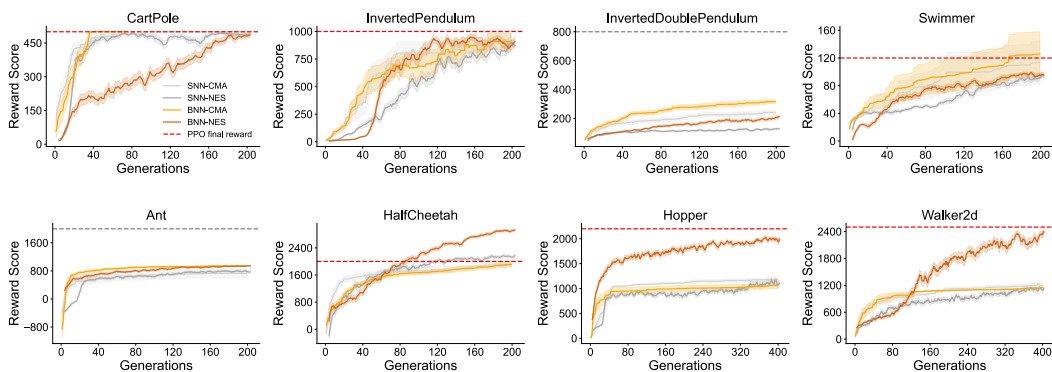

Figure 3: Performance of biological neurodynamic netwrok (BNN) and SNN on MuJoCo tasks. The dashed line indicates the state-of-the-art reference reward value achieved by PPO method reported in literature. Schulman et al. (2017)

**Model for synapse** In addition to neurons, the connections between them, *i.e.*, synapses, are crucial in biological neural networks. Synapses transmit electrical signals from presynaptic to postsynaptic neurons, it defines how signals fired by neurons affect the membrane voltage of a postsynaptic neuron. In the current work we choose to model two classes of synapses, AMPA and GABA, to describe the excitatory and inhibitory responses, respectively.

Synaptic connection strength, *i.e.*, synaptic weight, is the only trainable parameter in traditional neurodynamic models. Whereas in the real biological neural systems, the temporal order of signals from multiple synapses reaching the postsynaptic neuron is crucial to information encoding, hence this information processing fashion relies on synaptic delay. Mészáros et al. (2024) Therefore, we introduce **synaptic delay** as a novel parameter for better biological interoperability, which significantly improves the learning ability of our neurodynamic network.

**Neurodynamic network** Our network consists of 10 excitatory and 10 inhibitory neurons, and the number of neurons in the input and output group is dynamically adjusted according to the dimensionality of the task. Connections are established between all four groups within the network: Input, Excitatory (E), Inhibitory (I) and Output groups, via synapses. All the synapses starting from neurons in the Input, Output and E groups are excitatory, and synapses starting from the I group are inhibitory, as illustrated in figure 2A. This design follows Dale's Law and is consistent with the suggestions outlined in Burns (2021), who argues that incorporating heterogeneous neuron classes and biophysical constraints can enhance adaptability and interpretability in reinforcement learning models. Our model operationalizes these principles within an evolutionary-optimisation framework rather than Hebbian learning.

## 4.2 SIMULATION FRAMEWORK

**Evolutionary Algorithm** We introduce two popular evolutionary algorithms, CMA-ES Hansen et al. (2003) and NES. Wierstra et al. (2014) Among them, CMA-ES adaptively adjusts the mean and variance of the sampling distribution by calculating the covariance matrix, and NES updates the distribution by calculating the gradient through the Riemannian metric. For simple tasks CMA-ES can quickly find the convex regions on single-funnel energy landscapes characterised by low-dimensional spaces, whereas NES, through natural gradient climbing, does not rely on local second-order structures, and is therefore suitable for non-convex, multi-funnel complex landscapes, as illustrated in figure 2B.

A distributed setup for EAs is employed in the current work, which uses the island model as the base model. It is able to effectively balance the population diversity and global search capability, and is suitable for parallelisation in a distributed computing environment (DEC). By defining independent sub-populations and a periodic migration mechanism, it is able to execute in parallel on multiple

computing nodes. The introduction of the migration mechanism promotes the global search, and reduces the possibility of being trapped at local optima.

This design, inspired by population genetics, maintains partially isolated subpopulations that promote behavioural diversity and prevent premature convergence. Prior work (e.g., Salimans et al. (2017a)) did not rely on such mechanisms due to massive compute budgets, but recent studies highlight that under limited compute and highly non-convex fitness surfaces, island structures stabilise training by preserving exploration across disconnected regions of the landscape. We therefore include this mechanism to improve robustness of BNN optimisation.

**Dynamic simulation framework** The parallel neurodynamics training framework ENLARGE Qu et al. (2023) is employed. We mainly use the fine-grained network representation and hierarchical communication architecture in ENLARGE.

Unlike prior neuroevolution approaches such as Najarro Risi (2020), which optimize Hebbian plasticity rules within deep neural networks, our work directly evolves all parameters of a biologically grounded neurodynamic circuit—including synaptic delays, excitatory/inhibitory identity, and intrinsic dynamics. This shifts the optimisation target from plasticity parameters in ANNs to the full parameterization of a biophysically interpretable dynamical system.

## 5 EXPERIMENTS

We evaluate the above-described biological neurodynamic model, evolutionary algorithms and computational framework, and test them on multiple tasks from the MuJoCo environments. Todorov et al. (2012)

**Reinforcement learning task** Our BNN is assessed on a range of standard control tasks in a reinforcement learning environment, OpenAI Gym, Brockman et al. (2016) including some simple control task such as Cartpole, InvertedPendulum, InvertedDoublePendulum, and Swimmer, as well as several high-dimensional continuous control tasks from the MuJoCo environment, Todorov et al. (2012) including Ant, HalfCheetah, Hopper, and Walker2d. These tasks involve robotic locomotion control characterised by high-dimensional state spaces (ranging from 4 to 27 dimensions) and continuous action spaces (1 to 8 dimensions).

**Benchmark method** As benchmark algorithms, we used the Proximal Policy Optimization (PPO) Schulman et al. (2017) and the Trust Region Policy Optimization (TRPO) Schulman et al. (2015) implementation in RL-Baselines3-Zoo. Similarly, all tasks are implemented using the simulation environment provided by OpenAI Gym. Brockman et al. (2016) Both the policy and value functions are modelled as fully connected multi-layer perceptrons (MLPs) with two hidden layers of 64 units each, within tanh activations. Each task is trained for $5 \times 10^5$ steps, and performance is evaluated as the average return over 10 runs with different random seeds.

**Network parameter training** We employ the CMA-ES and NES fitness assessment strategy in the ENLARGE framework. $\lambda$ weights are sampled from a $[0, 1)$ random distribution as the initial individuals; then the fitness levels are assessed for individuals, and return to the main evolutionary algorithm; $\mu$ individuals with highest fitness values are selected to update the distribution for the sampler, and the best parameter and corresponding fitness value for each node is recorded; the best parameter is migrated across nodes every 50 generations. This procedure is repeated, until it reaches maximum generations.

**Deployment environment** All the training were performed on 40 nodes, each containing 2 Intel® Xeon® Platinum 8358@2.60GHz CPU, and 512GB RAM, with 10 tasks per node.

## 6 RESULTS

First, we benchmarked our BNN and a traditional SNN against the strong baseline performance achieved by MLP-PPO models reported in the literature. Schulman et al. (2017) Our intention is not to claim state-of-the-art results, but to evaluate the feasibility of evolutionary optimisation for

Table 2: Performance of BNN-EAs, SNN-EAs, MLP-PPO, and MLP-TRPO approaches in the MuJoCo tasks trained for the same number of optimisation steps. For each task, the top 2 performances are highlighted in bold.

| Methods | BNN-NES | BNN-CMA | SNN-NES | SNN-CMA | MLP-PPO | MLP-TRPO |
|---|---|---|---|---|---|---|
| **Environment Steps** | $5 \times 10^5$ | | | | | |
| **Model(Size)** | BNN(10,10) | | SNN(10,10) | | MLP(64,64) | |
| **Task** | Rewards | | | | | |
| CartPole | 485.1±8.9 | **500±0** | 490.6±5.7 | **500±0** | **500±0** | 496.4±2.8 |
| InvertedPendulum | 863.7 ± 40.3 | **1000.0±0.0** | 873.6±36.4 | **1000.0±0.0** | 217.8±99.6 | 842.3±33.9 |
| InvertedDoublePendulum | 203.5±11.5 | **348.3±13.4** | 127.1±5.9 | 289.1±22.0 | **4117.1±739.1** | 244.7±2.1 |
| Swimmer | 107.4±2.3 | 156.7±37.3 | 113.2±2.5 | 127.2±34.3 | **322.1±2.1** | **187.6±0.6** |
| Ant | **973.3±7.0** | **954.8±6.1** | 802.9±39.1 | 746.9±123.2 | 838.0 ± 68.5 | 367.8 ± 87.2 |
| HalfCheetah | **3187.4±36.8** | 2039.7±75.3 | 2265.2±25.9 | 1785.0±49.2 | **3818.6±120.3** | 2832.7±102.6 |
| Hopper | **2054.6±37.5** | 808.0±154.7 | 547.0±46.7 | 1190.2±40.3 | 780.3 ± 174.7 | **2153.3±53.9** |
| Walker2d | 2261.±90.6 | 1135.2±15.8 | 324.1±58.7 | 1044.4±5.2 | 1900.1±69.9 | **2427.9±105.3** |

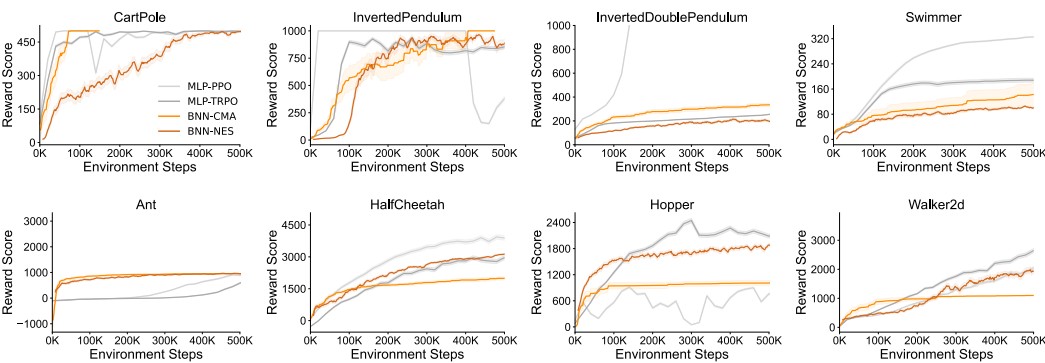

Figure 4: Training progress of BNN-CMA, BNN-NES, MLP-PPO and MLP-TRPO in 0.5 million steps.

biologically grounded networks. We trained BNN and SNN with two EAs, where the network size varied between 25 and 55 neurons, the most simple task CartPole requires 4 input neuron, 1 output neurons and 20 neurons in the E-I network, 25 neurons in total, and the most complicated task Ant requires 27 input neurons, 8 output neurons and 20 neurons in the E-I network, 55 neurons in total. The models were trained for up to 500 generations using both CMA-ES and NES, 1000 steps per generation. The results are illustrated in figure 3.

Results show that with substantially fewer training steps, our BNN achieves competitive performance for its parameter scale on most MuJoCo tasks. It consistently outperforms SNNs of the same scale and, on some tasks, achieves superior results with even fewer iterations. Furthermore, CMA-ES performs better on simpler tasks (top), whereas NES excels in more complex ones (bottom), consistent with the characteristics of their underlying energy landscapes (Figure 2B). This suggests the combination of the biological neurodynamic network and evolutionary algorithm (BNN-EA) is able to generate functional dynamic networks with effective task control capabilities in a much reduced model size, and with significantly fewer training steps.

To have a better insight into the training and convergence process, we further performed a comparison study between our BNN-EA approach, and SNN-EA, MLP gradient based approach. We trained a (64,64) MLP model with two hidden layers, with PPO and TRPO methods for the same $5 \times 10^5$ optimisation steps, and test them on MuJoCo control tasks. The results are summarised in table 2.

With the same number of optimisation steps, our BNN achieved comparable performance with MLP trained by PPO and TRPO. Furthermore, by inspecting the progression of the training, as illustrated in figure 4, suggests that our BNNs trained with EAs process similar learning dynamics as the mainstream reinforcement learning models and methods, and BNN-EAs has a higher sampling efficiency in the early stage of some complex tasks. Therefore the combination use of biological neurodynamic network and evolutionary algorithm training is a valid and feasible approach toward intelligence.

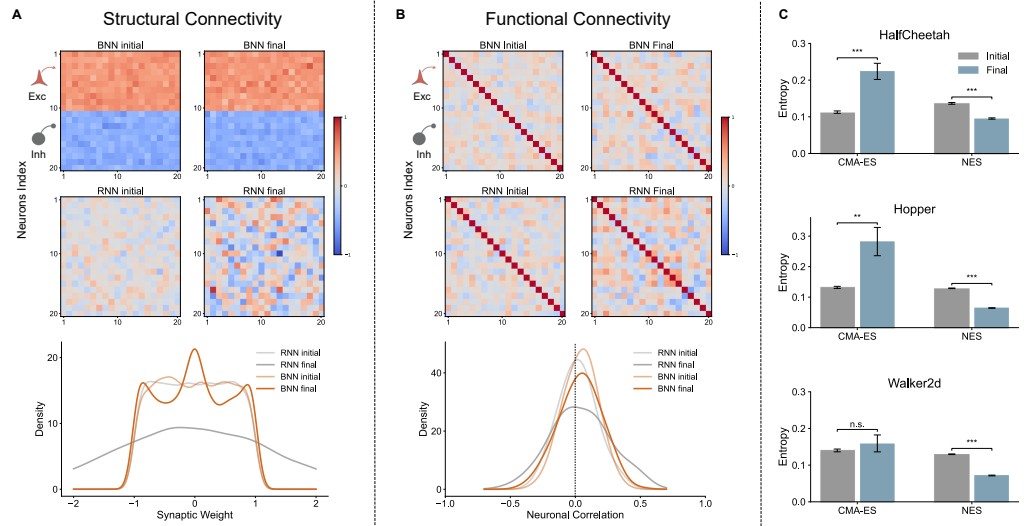

Figure 5: (A) The structural connection (SC) map of pre-trained and post-trained BNN (top) and RNN (middle). Colour describes the graded synapse, where red indicates excitatory, and blue indicates inhibitory. The distribution of the synapse weight is shown below. (B) The functional connection (FC) map of pre-trained and post-trained BNN (top) and RNN (middle). Colour describes the correlation, where red, and blue indicates one neuron has an excitatory, and inhibitory effect on the other neuron, respectively. The distribution of the neuron activity correlation is shown below.(C) The individual parameter distributions in BNN individuals of the CMA-ES and NES populations.**$p<0.01$, ***$p<0.001$.

However, for environments such as InvertedDoublePendulum and Swimmer, the BNN demonstrates noticeably weaker performance than the MLP baseline. These tasks require rapid, high-frequency corrective control (InvertedDoublePendulum) or smooth, coordinated periodic actuation (Swimmer). Such behaviours are known to be more challenging for dynamical spiking-inspired models with fixed synaptic delays and continuous postsynaptic currents, especially without task-specific tuning. As a proof-of-principle study, our goal is not to reach peak performance on every environment, but to demonstrate that biologically realistic neural dynamics can be effectively optimized in continuous-control tasks.

To further investigate the network structure of the evolved biological neurodynamic networks, and to examine the capability of the evolutionary algorithm in terms of producing certain network structures during training, we analysed our BNN by calculating the correlation between neuronal firing and the synapse weights of the network, for the pre-trained and post-trained networks, on CartPole task. The pre-trained network is defined as the initial random network, and the network subjected to 100 generations of EA optimisation. For comparison, we also trained a recurrent neural network (RNN) of the same size, *i.e.*, 20 neurons, using the same EA setup and did the same analysis as for the BNN. The results are presented in figure 5.

First, we calculate the structural connection, *i.e.*, synapse weights that connect neurons in the network. Our results suggest that BNN develops certain structures during optimisation using the evolutionary algorithm. For BNN the synapse weights are evenly distributed between -1 and 1 initially, and gradually shift to either -1 or 1 after training, suggesting some excitatory and inhibitory connections are established during evolutionary optimisation, and the network evolves some certain structures in response to external environment, *i.e.*, control tasks. For comparison, the RNN model did not evolve into any structures, and the synapse weights are even more random than the initial state. This observation proves that our BNN is able to evolve a structured functional network via evolutionary optimisation. Similarly, we also analyse the functional correlation between the neurons, which describes whether one neuron has an excitatory (positively correlated) or inhibitory (negatively correlated) effect on the other neuron, thus providing insight into the internal dynamics of the network. The result suggests our BNN develops some correlations between neurons, as more

neuronal correlations are shifted away from zero, whereas the neuronal correlation distribution for RNN remains unchanged, suggesting no certain structures were formed during training for RNN.

Figure 5C shows the parameter distribution of individuals within an EA population, measured by KL divergence, which served as an analogue to entropy in statistical thermodynamics. The parameter distribution provides an ideal indication of the parameter, or gene diversity of populations. Our results suggest that the CMA-ES algorithm tend to increase the parameter diversity, whereas for NES algorithm all individuals tend to converge to the same region in the parameter space. Populations with better gene diversity are likely to preform well upon task switching, and therefore avoid catastrophic forgotten, thus presenting a promising pathway toward continual learning.

## 7    DISCUSSION

The above described work shows that the combination use of our proposed biological neurodynamic networks and the evolutionary algorithms is capable of producing functional networks for control tasks with competitive performance given its significantly reduced model size, though not necessarily matching state-of-the-art gradient-based RL methods, and within a significantly fewer training steps. Furthermore, as a proof-of-principle study we demonstrate that the BNN can evolve certain network structures during evolution. Our BNN framework therefore provides a biologically grounded and interpretable route to intelligence, standing as a compelling parallel to mainstream reinforcement learning methodologies.

Beyond establishing performance feasibility, this framework enables controlled investigation of how biophysical mechanisms—such as synaptic delay heterogeneity, E–I population structure, and dynamical stability—shape the emergence of functional behaviour in embodied tasks. Because these mechanisms are difficult to study using gradient-based methods and rarely appear jointly in prior evolutionary studies, our approach provides a testbed for quantifying the computational roles of biological dynamics in reinforcement-learning-like settings.

**Limitations and future works**    Our current work focuses on small networks with only 20 neurons, whereas real biological brains typically contain millions or even billions of neurons. We already achieved the state-of-the-art performance in most MuJoCo control tasks, and began to observe some certain network structures from evolution. Therefore we anticipate that by scaling up the BNN the control task performance will further increase, and more biologically relevant and interpretable structures will emerge, which would benefit both the neuroscience and reinforcement learning community. Furthermore, the initial structure of our BNN is relatively simple, further development of the model can introduce extra layers or *a priori* empirical structures to improve the performance and bio-interpretability. In addition, the biologically realistic nature of BNN and EA allows us to further investigate the environment adpation behaviour of a population of individal networks using the analougue of thermodynamic entropy. Owing to the population based nature of EAs, we will investigate the interplay between parameter (gene) diversity and task (environment) adaption capabilities in further studies.

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

# Appendix

## A Biological Neurodynamic Models

**Model for neuron** Neurodynamic models require neuron models to describe the dynamics of neurons, *i.e.*, signal firing. Multiple models for neurons have been proposed; among them, the Leaky Integrate-and-Fire (LIF) neuron Gerstner et al. (2014) has been proved to be a biologically realistic and computationally feasible one. The state of LIF neurons can be expressed as:

$$C_m \frac{dv(t)}{dt} = -\frac{C_m}{\tau_m}[v(t) - v(rest)] + i_e(t) + i_i(t) + I_{offset} + I_{injection}(t), \tag{1}$$

$$\text{if } V(t) > V_{th}, V_t \leftarrow V_{reset}, \tag{2}$$

where $C_m$ is the neuron capacity, $v(t)$ is the membrane voltage, $\tau_m$ is the membrane time constant, $v(rest)$ is the resistant voltage, $i_e(t)$ and $i_i(t)$ are the excitatory and inhibitory input currents, respectively, and $I_{offset}$ and $I_{injection}(t)$ are the constant / noisy input currents. If the membrane voltage $v(t)$ reaches the threshold $V_{th}$, the neuron will issue a signal, and $v(t)$ will be held at $v(rest)$ for a refractory period $\tau_{ref}$. Once this refractory period ends, the neuron follows this expression until it issues the next signal again.

**Model for synapse** In addition to neurons, the connections between them, *i.e.*, synapses, are crucial in biological neural networks. Gerstner et al. (2014) Synapses transmit electrical signals from presynaptic to postsynaptic neurons, it defines how signals fired by neurons affect the membrane voltage of a postsynaptic neuron. In the model for synapses, it has the impulse signal in presynaptic neurons as the input, and the postsynaptic current as the output. The postsynaptic current (PSC) can be expressed as:

$$I_{syn} = g(t)(V_{post} - E_{syn}), \tag{3}$$

where $I_{syn}$ is the postsynaptic current, $V_{post}$ is the postsynaptic voltage, $E_{syn}$ is the reversal potential of the synapse; its value determines whether a synapse is either excitatory or inhibitory, and $g$ is the conductance on synapses.

In phenomenological models, *i.e.*, synapse is modelled based on the postsynaptic current, rather than the dynamical features of the ion channels, the time-dependence of $g(t)$ can be expressed as:

$$g_t = \bar{g} \sum_{t^{(f)}} s(t - t^{(f)}), \tag{4}$$

where $\bar{g}$ is a constant describing the synapse weight, $t^{(f)}$ denotes the moment that the signal arrives at the synapse, $s(t - t^{(f)})$ is a time-dependent function that describes the effect of signal firing on $g_t$. Using an exponential decay to model the synapses, we obtain:

$$s(t) = e^{-t/\tau} H(t), \tag{5}$$

where $\tau$ is a temporal constant, and $H(t)$ is the heaviside function. We introduce synaptic delay as a new parameter for better biological interoperability, the $g(t)$ differential equation of $g_t$, with respect to the current time is therefore:

$$\frac{dg}{dt} = -\frac{g}{\tau} + \bar{g} \sum_{t^{(f)},k} \delta(t - t^{(f)} - d^k). \tag{6}$$

Where $\delta$ function is zero anywhere but $t = 0$, and $t$ is the current time, $d^k$ is the transmission delay corresponding to the $k^{th}$ synapse.

Unlike most ANN and SNN benchmarks that assume uniform synaptic delays, biological synapses exhibit heterogeneous conduction and transmission latencies. Such delay variability has been shown to create rich temporal structures—including polychronous firing assemblies (Izhikevich (2006))—which increase memory capacity and extend the temporal credit-assignment horizon. Optimising delays therefore expands the dynamical repertoire of the BNN beyond what can be achieved using weights alone.

The parameters for neuron and synapse models employed in the work described in this paper are summarised in table 3.

Table 3: Parameters for the Biological Neurodynamic Model.

| Parameter | Value | Unit |
|---|---|---|
| Resting potential ($V_{\text{rest}}$) | -60 | mV |
| Reset potential ($V_{\text{reset}}$) | -60 | mV |
| Initial membrane potential ($V_0$) | -55 | mV |
| Threshold potential ($V_{\text{th}}$) | -50 | mV |
| Membrane capacitance ($C_m$) | 1 | pF |
| Membrane time constant ($\tau_m$) | 20 | ms |
| Synaptic time constant ($\tau_{\text{syn}}$) | 5 | ms |

## B  BIOLOGICAL NEURODYNAMIC NETWORKS

**Excitatory–Inhibitory Balance Models**  Excitatory–inhibitory (E–I) balance Vreeswijk & Sompolinsky (1996) is a hallmark of cortical microcircuitry, whereby excitatory and inhibitory synaptic inputs dynamically counterbalance each other to maintain stable yet flexible neural activity. This principle, rooted in electrophysiological recordings from the neocortex and hippocampus, has inspired a class of computational models that preserve this balance at both the single-neuron and population level. These E–I balanced networks typically consist of spiking neurons or rate-based models partitioned into excitatory and inhibitory subpopulations, with constrained synaptic weights (e.g., Dale's law Strata & Harvey (1999)) and recurrent connectivity that gives rise to asynchronous irregular activity, criticality, and efficient coding. The E–I architecture has been shown to support robust computations, including working memory, pattern decorrelation, and gain control, by leveraging biologically plausible dynamics rather than task-specific training. Notably, such models often operate without supervised gradient-based optimization, instead relying on biologically motivated plasticity rules or structured connectivity to enable function.

The parameters for the biological neurodynamic network (BNN) employed in the work described in this paper are summarised in table 4.

Table 4: Parameters for the Biological Neurodynamic Model.

| Parameter | Value | Unit |
|---|---|---|
| Simulation duration | 100 | timestep |
| Simulation step size | 0.1 | ms |
| Number of neurons | 10E + 10I + number of input&output | - |
| Learnable Params | synaptic weights and delays, noisy input | - |

## C  SPIKING NEURAL NETWORKS

Spiking neural networks (SNNs) Maass (1997); Maass & Markram (2004) emulate the dynamics of real neurons by encoding and transmitting information via discrete spikes. Neurons in SNNs integrate synaptic inputs over time and emit spikes when membrane potentials exceed threshold, introducing an inherent temporal dimension and nonlinearity that makes them both biologically plausible and computationally distinct. Unlike MLPs, where activations are continuously differentiable, SNNs rely on non-differentiable events, posing challenges for standard optimization methods

and prompting the development of surrogate gradients, spike-based learning rules, and biologically inspired plasticity mechanisms. Furthermore, the asynchronous and sparse firing nature of SNNs facilitates event-driven computation, offering potential gains in energy efficiency when deployed on neuromorphic hardware. Thus, while MLPs excel in data-rich, high-throughput regimes with dense numerical representations, SNNs offer a promising alternative for real-time, low-power, and biologically grounded computation, particularly in scenarios demanding temporal precision and structural interpretability.

## D  ARTIFICIAL NEURAL NETWORKS

Alongside our proposed biological neurodynamic networks (BNNs), the following artifical neural networks LeCun et al. (2015) are considered in the work described in this papar.

**Multilayer perceptrons**  Multilayer perceptrons (MLPs) Hornik et al. (1989) are feedforward neural networks that compute deterministic, continuous-valued transformations through stacked layers of weighted summations and pointwise nonlinearities. As universal function approximators, MLPs form the backbone of modern deep learning, enabling supervised and unsupervised learning across vision, language, and control domains. Their success is rooted in architectural simplicity, differentiability, and compatibility with efficient gradient-based training methods such as backpropagation. However, MLPs operate in discrete time with synchronous updates and dense, analog activations—features that stand in sharp contrast to the event-driven, temporally sparse computations observed in biological neural circuits.

**Recurrent Neural Networks**  Recurrent neural networks (RNNs) Rumelhart et al. (1986) are a foundational class of artificial neural models developed to handle sequential and temporally structured data. Unlike feedforward architectures, RNNs maintain internal state through recurrent connections, allowing them to capture dynamic dependencies across time. Variants such as Long Short-Term Memory (LSTM) Hochreiter & Schmidhuber (1997) networks and Gated Recurrent Units (GRUs) were introduced to overcome vanishing gradient issues and have since become standard tools in natural language processing, time series forecasting, and reinforcement learning. While RNNs exhibit impressive performance in engineering tasks, they lack constraints from biological connectivity and synaptic dynamics. Their recurrent activity arises from parameterized weight matrices and nonlinear activations, rather than the structured E–I coupling observed in cortical networks. As such, RNNs prioritize trainability and function approximation over mechanistic interpretability. Efforts to bridge this gap have emerged through hybrid models, such as balanced RNNs or spiking RNNs with biologically inspired constraints, but a fundamental divergence remains: E–I balance models are grounded in the physical dynamics of neural tissue, while RNNs abstract away biological detail to maximize learning flexibility.

The key differences between MLP, SNN and our BNN are summarised in table 5.

Table 5: Comparison of different network architectures.

| Network Architecture | Artificial Neural Network | Spiking Neural Network | Biological Neurodynamic Network |
|---|---|---|---|
| Basic Unit | Weighted sum + nonlinear activation | integrate-and-fire | LIF + Synaptic dynamics |
| Neruonal Dynamics | Static | Temporal integration | Membrane potential + conductance dynamics |
| Information Encoding | Continuous values | Spike trains | Continuous dynamical state |
| Computational Efficiency | High (GPU friendly) | Medium (requires event-driven framework) | Low to Medium (complex simulation) |
| Application Domain | Image recognition, NLP | Brain-inspired computation | Brain modeling, biophysical simulations |

## E  PARALLEL SIMULATION FRAMEWORK

The parallel neurodynamics simulation framework ENLARGE Qu et al. (2023) is employed. We mainly use the fine-grained network representation and hierarchical communication architecture in ENLARGE.

Fine-grained impulse neural network represents and stores neuron and synapse parameters, and network topology separately. The neuron and synapse parameters are firstly separated and grouped according to the characteristics of the computational units and distributed memory. The communication paths are then optimised by the segmentation algorithm to reduce the redundant commu-

nication between clusters. Compressed sparse row (CSR) analogy is used for network topology information to describe the connection information of each individual neurons and synapses. For the fine-grained pulsed neural network representation, the delay information is integrated into the CSR representation to form a compact network representation, which dramatically saves the storage requirement.

The hierarchical communication architecture divides cluster communication into three levels: process level, intra-node level and inter-node level. It provides two main modules: the converter module and the communication module. The former locates in the process, which queries the firing neuron ID from the firing list, and converts it into the corresponding shadow neuron ID; it also rearranges the shadow neuron IDs according to their destinations. The latter handles most of the communication and synchronisation. All core computations are performed in the process, the firing list is stored and accessed at each computation phases for data sharing. The synchronisation occurs only at the inter-node level. As the global time must be synchronised within each time step, the hierarchical communication architecture transmits the data in a blocking manner in order to reduce the cost. This blocking communication process can also be used as a synchronisation signal, instead of the actual synchronisation process of the global time.

## F  PARALLEL EVOLUTIONARY ALGORITHM

We considered two implementations of the evolutionary algorithms, CMA-ES Hansen et al. (2003) and NES Wierstra et al. (2014). The pusedocode is presented in **Algorithm** F and **Algorithm** F.

Parallel CMA-ES Algorithm **Input:** Initial network parameters $\theta_0$, noise standard deviation $\sigma$.
**Output:** Optimal network weights $W$.
**Initialise:** $m$ nodes, $n$ tasks.
generation < generation limit each nodes i=1 to m  each tasks j=1 to n Generate a polpulation of network parameters using CMA-ES sampler;
Parallel compute fitness score $F_j$ and return to CMA-ES;
Update CMA-ES sampler's distribution;
Record best parameter $\sigma_i$, best score $F_i$ of all tasks;
generation = migrate generation Exchange best parameter $\theta_i$, best score $F_i$ to every other nodes;
Save network weights with highest score $W_{best}$.

Parallel NES Algorithm **Input:** Initial network parameters $\theta_0$, noise standard deviation $\sigma$.
**Output:** Optimal network weights $W$.
**Initialise:** $m$ nodes, $n$ tasks.
generation < generation limit each nodes i=1 to m  each tasks j=1 to n Generate a polpulation of network parameters using NES sampler;
Parallel compute fitness score $F_j$ and return to NES;
Update NES sampler's distribution;
Record best parameter $\sigma_i$, best score $F_i$ of all tasks;
generation = migrate generation Exchange best parameter $\theta_i$, best score $F_i$ to every other nodes;
Save network weights with highest score $W_{best}$.

The parameters for evolutionary algorithms employed in the work presented in this paper are listed in table 6.

Table 6: Parameters of Evolutionary Algorithm.

| Parameter | Value |
|---|---|
| **Population size ($\lambda$)** | 10 |
| **Parent size ($\mu$)** | $\lambda/2$ |
| **Initial step size ($\sigma$)** | 0.5 |
| **Initial mean** | Uniform$([0, 1)^n)$ |
| **Max generation** | 500 |

## G  REINFORCEMENT LEARNING TRAINING ALGORITHMS

We consider two RL training techniques, PPO Schulman et al. (2017) and TRPO Schulman et al. (2015) in the current work. The parameters during network training presented in the work described in this paper are listed in table 7.

Table 7: PPO and TRPO parameters used for Mujoco tasks.

| Parameter | Value | Description |
|---|---|---|
| Policy Net | MLP | 2 layers with 64 units each |
| Learning rate | $3 \times 10^{-4}$ | Adam optimizer |
| Discount factor ($\gamma$) | 0.99 | Reward discounting |
| GAE lambda ($\lambda$) | 0.95 | For Generalized Advantage Estimation |
| Batch size | 64 | Mini-batch size for updates |
| Rollout length | 2048 | Timesteps per batch |
| Epochs per update | 10 | Number of gradient steps per batch |

## H  EXPERIMENTS

**CartPole**  CartPole Barto et al. (1983) is a 2-dimensional control task in which the intelligent agent needs to keep a pole balanced upright by applying discrete left and right thrusts. The task, which is widely regarded as a standard test for assessing the responsiveness and stability of a controller. The state space is four-dimensional, comprising the cart's position and velocity, as well as the pole's angle and angular velocity. The 2-dimensional action space consists of two discrete actions representing forces applied to the left or right. The reward of the intelligent agent is determined by the positive feedback gained from successfully maintaining the equilibrium state at each time step, and the task continues until the pole tilt angle or position exceeds a threshold value.

**MuJuCo Tasks**  The MuJoCo tasks Todorov et al. (2012) we chose are mainly robot motion control problems with a high-dimensional state space (8-27 dimensions) and a continuous action space (1-8 dimensions), where the state contains information about the position, velocity, angle, and angular velocity of the body parts, and the action represents the control signals applied to the actuated joints. The reward functions primarily based on the the distance travelled along a specific axis, complemented by an energy penalty term and balance constraints. Each task is set as a finite time trajectory of 1000 steps.

