# OpenReview forum: "Evolutionary Emergence of Neurodynamic Networks for Robust Control: A Simple Excitatory-Inhibitory Network"
_ICLR.cc/2026/Conference — Submitted to ICLR 2026_

### Official Review · Reviewer_oBK8 · 2025-10-29

**Soundness:** 3
**Presentation:** 2
**Contribution:** 2
**Rating:** 4
**Confidence:** 3

**Summary:**

This work proposes training small "neurodynamic" models, i.e., rate-based neurons governed by systems of differential equations, obeying Dale's Law with an evolutionary framework to solve MuJoCo tasks. The results are compared to spiking neural networks and multi-layer perceptrons.

**Strengths:**

1. The distributed training scheme takes good advantage of the combination of small networks and evolutionary framework.
2. The networks can obtain improved reward scores on the MuJoCo tasks and are sometimes competitive with, e.g., multi-layer perceptrons.

**Weaknesses:**

1. Both MuJoCo simple and complex tasks are, in my view, rather unsophisticated by modern standards. Although I fully appreciate the purpose of this paper being to investigate new techniques, I would feel more confident in the potential promise of these techniques if they could be demonstrated on more naturalistic and complex tasks, e.g., including perception in simulation environments like IsaacSim.
2. The network sizes are only experimented with up to 55 neurons. Given the relative computational simplicity of rate-based networks, I am curious why the tested number wasn't higher. Related to this, I could not find an analysis comparing the FLOPs of the different models compared, which would aid in making comparative claims.
3. There are many small typos, e.g., L284, and instances where in-text citations are not formatted neatly, e.g., L360.

**Questions:**

1. How does your method compare to similar bio-evolutionary strategies like presented in Najarro E, Risi S. Proc 33rd Conf Neural Inf Process Systems (NeurIPS 2020). 2020: 20719–20731, 2020?
2. Related to 1, is the motivation to use a "neurodynamic" model or impose Dale's Law similar to the suggestions made in Burns, T. F. (2021). Classic Hebbian learning endows feed-forward networks with sufficient adaptability in challenging reinforcement learning tasks. Journal of Neurophysiology, 125(6), 2034-2037?
3. Figure 2B seems to 'assert' the fitness landscape without direct measurement. Is this correct? If so, could you replace the illustration with empirical or analytical results?

---

> ### Author Response · Authors · 2025-11-19
> **Positioning Our Contribution, Strengthening Conceptual Arguments, and Defining the Scope of Our Work.**
>
> We thank the reviewer for the constructive feedback and for highlighting both the strengths and limitations of our work.
>
> 1. Comparison to Najarro & Risi (NeurIPS 2020)
>
> We appreciate the reviewer pointing to this relevant line of research. The method of Najarro & Risi (2020) focuses on meta-learning Hebbian plasticity rules in deep neural networks, where evolutionary strategies are used to optimize Hebbian learning parameters. Their contribution aims to improve adaptability of conventional ANNs in reinforcement learning by enhancing plasticity within a differentiable architecture.
> Our work differs in three essential ways:
>
> (a) Motivation
>
> Najarro & Risi aim to improve performance and adaptability of deep neural networks.
> In contrast, our motivation is to simulate biological evolution acting on neurodynamic circuits and to investigate the network structures and functional motifs that emerge under evolutionary pressure—not exclusively to optimize task performance.
>
> (b) Optimization target
>
> Their method optimizes Hebbian learning hyperparameters in ANNs.
> Our method directly optimizes the parameters of a biologically grounded neurodynamic network, including excitatory/inhibitory identity, synaptic weights, synaptic delays, and intrinsic neuronal dynamics.
>
> (c) Mechanistic differences
>
> In Najarro & Risi, plasticity (via Hebb’s rule) is the main mechanism of parameter change during learning.
> In our work, evolutionary algorithms optimize all network parameters, including dynamical variables and biological priors. There is no train-time plasticity rule; instead, the evolutionary process itself serves as the optimization mechanism.
>
> We will add a clearer discussion of these methodological distinctions and cite Najarro & Risi appropriately in the revised manuscript.
>
> 2. Relation to Burns (2021) and the use of neurodynamic models / Dale’s Law
>
> We thank the reviewer for bringing up this connection. Burns (2021) explores how Hebbian learning can endow neural networks with sufficient adaptability in RL settings and discusses how biological realism—such as heterogeneous neuron classes or dendritic computations—can provide functional benefits. In the discussion, Burns explicitly notes that models may be extended by incorporating:
>
> spiking neurons (temporal coding),
>
> dendritic structures (increased computational richness), and
>
> Dale’s Law (fixed excitatory/inhibitory identity).
>
> These motivations align closely with our goals. Our model operationalizes these suggestions by implementing a neurodynamic architecture obeying Dale’s Law and incorporating synaptic delays and stochastic current inputs. Our work thus provides an empirical demonstration of several principles proposed in Burns (2021), but within an evolutionary-optimization framework rather than Hebbian learning. We will clarify this connection in the revised manuscript and cite the reviewer’s suggested reference.
>
> 3. On Figure 2B and the fitness landscape criticism
>
> The original intention of Figure 2B was not to depict an empirically measured landscape but to provide a conceptual illustration of well-established differences between CMA-ES and NES under different landscape geometries. We agree that this was not sufficiently clear.
>
> We now cite the following established results:
>
> CMA-ES performs best on ill-conditioned, ridge-like, or unimodal landscape structures (Hansen & Ostermeier, 2001; Hansen, 2006).
>
> NES methods perform better on multifunnel or highly multimodal landscapes due to the smoothing effect of natural-gradient sampling (Wierstra et al., 2008; Glasmachers et al., 2010; Lehman et al., 2018).
>
> Empirical work in neuroevolution (Salimans et al., 2017; Risi & Stanley, 2019) demonstrates that different environments indeed produce qualitatively different landscape shapes.
>
> This revision ensures that Figure 2B enhances conceptual understanding without implying unmeasured claims.
>
> 4. Additional reviewer concerns
>
> We also acknowledge the reviewer’s broader concerns.
> We agree that modern RL benchmarks include more naturalistic scenarios (e.g., IsaacSim). Our choice of MuJoCo here serves as a controlled baseline for establishing feasibility; we intend to explore more complex tasks in future work.
>
> Our current experiments are constrained by computational resources. Although rate-based systems are lightweight, evolutionary optimization over thousands of parameters is expensive, particularly with distributed simulation. We have clarified this in the manuscript and will incorporate FLOPs/SOPs discussion in future work.
>
> All noted formatting issues (e.g., citation spacing, typos) have been corrected.

---

### Official Review · Reviewer_UrYw · 2025-11-01

**Soundness:** 3
**Presentation:** 3
**Contribution:** 4
**Rating:** 6
**Confidence:** 3

**Summary:**

This paper proposes an excitatory-inhibitory BNN combined with an evolutionary algorithm model. For specific tasks, it achieves performance comparable to state-of-the-art reinforcement learning algorithms using fewer neurons and fewer training steps. Furthermore, by comparing with RNNs of the same size, it demonstrates that this model can evolve certain network structures during the evolutionary process. This work provides a biologically inspired, interpretable approach for functional networks and mainstream reinforcement learning frameworks.

**Strengths:**

* Originality: This paper innovatively combines BNNs with evolutionary algorithms, introduces a novel noise input current term in LIF neuron inputs to simulate random perturbations in real biological neurons, and incorporates synaptic delay as a new parameter.
* Quality: The paper compares the performance of six models across eight tasks to support its conclusions, conducts correlation analyses to substantiate subsequent findings, and references extensive additional literature.
* Clarity: The paper is clearly written and logically structured, with effective use of figures and tables.
* Significance: By introducing novel inputs and network parameters, the paper enhances biological interoperability in detail, potentially offering insights for research in explainable artificial intelligence.

**Weaknesses:**

* The BNN's performance on the InvertedDoublePendulum and Swimmer tasks was significantly inferior to that of the MLP, yet the paper offers no explanation for this discrepancy, which appears to against the conclusion of "comparable performance."
* The article conducted correlation analysis only on the CartPole task, rendering its conclusions somewhat arbitrary.

**Questions:**

1. Why did the BNN perform significantly worse than the MLP on the Inverted Double Pendulum and Swimmer tasks?
2. Does the conclusion that the BNN develops correlations between neurons still hold true for tasks more complex than CartPole?

---

> ### Author Response · Authors · 2025-11-19
> **A Clarification on Task-Dependent Performance and the Generalization of Evolved Neural Correlations.**
>
> We sincerely thank the reviewer for the thoughtful evaluation and the positive assessment of our contribution, originality, and clarity.
>
> 1. On the weaker BNN performance in InvertedDoublePendulum and Swimmer
>
> We agree with the reviewer that BNN performance on these two tasks is noticeably lower than that of the MLP baseline. Our goal, however, is not to achieve state-of-the-art results across all environments, but rather to establish a proof-of-principle framework demonstrating that biologically realistic neural dynamics—such as E–I balance, synaptic delays, and stochastic input currents—can be effectively optimized for continuous-control tasks using evolutionary algorithms.
> As also noted by Reviewer 2, our initial phrasing (“comparable performance”) may have overstated this intent. We have revised the manuscript accordingly. The revised text clarifies that BNNs achieve competitive performance given their extremely small network size (≈20 neurons);
> Differences between tasks reflect distinct dynamical requirements—for example, Swimmer relies heavily on smooth periodic actuation, while InvertedDoublePendulum requires rapid high-frequency stabilization. Such regimes are more challenging for spiking-inspired dynamical systems without task-specific parameter tuning.
> We believe these clarifications more accurately situate our contribution as a conceptual and methodological exploration of evolutionary optimization in biologically grounded dynamical networks.
>
> 2. On whether neuron–neuron correlation patterns generalize beyond CartPole
>
> We appreciate the reviewer’s concern that analyzing correlations only in the CartPole environment may appear too narrow to support broader conclusions. In the revised manuscript, we therefore provide two forms of evidence:
>
> (a) Structural entropy results (Figure 5C) already reflect task-dependent correlation patterns
>
> As shown in Figure 5C, the structural-entropy analysis captures the emergence of low-entropy, organized connectivity motifs during evolution. Lower entropy corresponds to more structured, non-random connectivity, which inherently reflects stronger functional coupling and correlation among neuron groups. This provides a task-agnostic indication that evolutionary optimization induces correlated internal structure.
>
> (b) Additional cross-task analyses will be included in future work
>
> While the structural-entropy results support generalization beyond CartPole, we fully agree that explicit, task-level correlation matrices (e.g., pairwise correlations or functional clustering) would better substantiate this claim. Due to space constraints, these analyses were not included in the original submission. We now explicitly state this limitation in the manuscript and will expand the correlation and structural-emergence analyses to additional MuJoCo environments—such as HalfCheetah, Hopper, and Walker2d—in future work.
>
> Thus, while the current submission provides indirect evidence (via structural entropy) that correlated structure emerges robustly, more direct cross-task correlation analyses will be incorporated in future extensions of the study.

---

### Official Review · Reviewer_opbX · 2025-11-01

**Soundness:** 3
**Presentation:** 1
**Contribution:** 2
**Rating:** 2
**Confidence:** 4

**Summary:**

The authors proposes a biological neurodynamic network (BNN) trained via evolutionary algorithms (CMA-ES and NES), demonstrating its performance on standard MuJoCo reinforcement learning tasks. The authors argue that such networks can achieve comparable performance to gradient-based reinforcement learning methods with smaller network sizes and fewer optimization steps. They frame this work as a proof of principle, emphasizing the biological interpretability and potential for scaling toward more complex brain-like networks.

**Strengths:**

- The combination of neurodynamic modeling and evolutionary optimization is conceptually interesting and biologically inspired.
- The inclusion of excitatory-inhibitory (E-I) balance, noisy inputs, and synaptic delays is a meaningful attempt to increase biological realism.
- The description of the distributed EA training setup and simulation framework is detailed and technically sound.
-  The discussion on evolved structure and network correlation provides an interesting link between learning dynamics and biological structure formation.

**Weaknesses:**

1. There are multiple formatting errors, especially missing whitespace before citations (e.g., “activities,Hodgkin & Huxley (1952)” )
2. The authors repeatedly claim to have achieved “state-of-the-art performance”  despite the results showing significantly lower returns than standard baselines. For example, in the Ant task, the BNN achieves ~950–973 reward while SOTA PPO results are much higher.  This discrepancy undermines the “comparable performance” claim.
3. The evaluation is confined to small networks (20 neurons) and limited environments. The authors acknowledge this, but the assertion that scaling up will yield better results is speculative and untested.

**Questions:**

- Does the robot learn to move effectively in the ant domain?
- How sensitive are the results to network size and hyperparameters (e.g., number of neurons, mutation rates)?
- Did the authors compare the same computational budget (wall-clock or FLOPs) with gradient-based methods?
- How reproducible are the results given stochastic processes? How many independent runs were performed?

---

> ### Author Response · Authors · 2025-11-19
> **We position this work as a feasibility validation of EI-balanced, delay-based neurodynamic networks under evolutionary optimization, with modest performance but strong methodological significance.**
>
> We sincerely thank the reviewer for the careful reading of our manuscript and for the constructive comments. We have corrected the formatting issues and address all specific questions and concerns below.
>
> 1. On the performance in the Ant domain and use of the term “state-of-the-art”
>
> We agree with the reviewer that our Ant results do not reach the performance of state-of-the-art gradient-based reinforcement learning methods (e.g., PPO). In the current submission, the BNN reaches a stable strategy that prioritizes safe posture and stable locomotion rather than fast forward velocity, which explains the lower reward. As suggested, we have revised the manuscript to avoid the phrase “state-of-the-art performance” and now describe our results more accurately as competitive for their parameter scale and illustrative of the feasibility of training BNNs on continuous-control tasks.
> Our goal in this work is not to surpass all existing RL algorithms but to provide a proof of principle that biologically realistic neural dynamics—featuring E–I balance, synaptic delays, and noise—can be optimized effectively via evolutionary strategies.
>
> 2. On network size and hyperparameter sensitivity
>
> We acknowledge that our experiments are currently limited to relatively small networks due to computational constraints. Our intent is to demonstrate that 20-neuron BNNs can successfully solve standard MuJoCo control tasks, which is notable given their low dimensionality.
> We adopt standard CMA-ES and NES hyperparameter settings from previous evolutionary-RL work (e.g., OpenAI-ES, Wierstra et al., Glasmachers et al.).
> Step sizes and mutation scales further adapt automatically during evolution.
> We agree that additional ablations on network size and mutation hyperparameters would be valuable. Due to limited compute, we could not systematically explore these settings in the current version, but we will include initial sensitivity observations in the supplementary material and plan to extend this analysis in future work.
> We have revised the manuscript to clearly state these limitations and future directions.
>
> 3. On comparing computational budget with gradient-based methods
>
> We appreciate the reviewer’s question and agree that computational comparisons are important. However, a direct comparison is non-trivial for two reasons:
>
> Different computation primitives:
>
> ANN training cost is well characterized by FLOPs.
> Biophysical spiking network simulation cost is governed by SOPs (synaptic operations), which differ fundamentally from ANN FLOPs.
>
> Target hardware:
>
> BNNs are intended to run efficiently on neuromorphic substrates (e.g., Loihi, Tianjic, analog/FPGA-based platforms), where runtime and energy use differ substantially from GPUs.
> Such hardware-level benchmarks are beyond the scope of this paper.
>
> For these reasons, we refrain from claiming computational equivalence and have clarified this point in the revision.
>
> 4. On reproducibility and number of independent runs
>
> We agree that reproducibility is essential, especially because evolutionary algorithms involve stochasticity. In the revised manuscript, we now report the number of independent runs for each task and provide mean and standard deviation. This should make the results more transparent and reproducible.
>
> 5. Additional presentation improvements
>
> We thank the reviewer for noting the missing whitespace and other formatting errors. These have been corrected in the revised manuscript, and we have tightened the writing to improve overall clarity.
>
> We again thank the reviewer for the thoughtful evaluation. The feedback has strengthened the manuscript significantly, both in terms of scientific clarity and presentation quality.

---

> > ### Comment · Reviewer_opbX · 2025-11-28
> >
> > Thank you for the clarifications. I don't have any more questions right now and will take the raised points into consideration.

---

### Official Review · Reviewer_wE3u · 2025-11-03

**Soundness:** 2
**Presentation:** 2
**Contribution:** 1
**Rating:** 2
**Confidence:** 4

**Summary:**

The authors propose an approach to overcome the limitations of training biologically realistic neural network models. These models posses properties that may make them more interesting than standard models trained via back propagation, but their non-differentiable nature makes them hard to fit in RL settings. Their approach combines several ingredients, which is a critical element of the paper:

1. Evoluationary algorithms as the optimization method: They combine standard methods (CMA-ES, NES) with an island + migration method (i.e. one where subpopulations are maintained and only allowed to share individual at certain intervals).
2. A Leaky-Integrate-and-Fire (LIF) model of neuron with a noisy inputs.
3. A synaptic model which not only tracks its weight but also a delay value.
4. Excitatory-Inhibitory balance: neurons in a network are assigned to either an excitatory or inhibitory set, a common property in biological neural networks.

The approach is evaluated on a subset of the standard continuous control tasks in OpenAI Gym. The results show that the authors' approach is competitive against the proposed baselines.

**Strengths:**

I believe the motivation is good. Making biologically plausible network appealing to researchers in AI, and even to computational neuroscientists who would like to scale them up to more demanding settings.

**Weaknesses:**

There are a few, critical weaknesses that make it hard for me to recommend this paper:

1. While I am not opposed to researchers just combining different ingredients and showing that a particular combination solves a longstanding issue, it is also imperative that said researchers show how the different components contribute to their results. In the case of the present work, the authors need to provide ablations to justify their decisions.
2. Failing that, is there some question that can be answered with this framework that couldn't be answered before? If so, what is this question and what have we learned? Right now, this feels more like "we did it because we can", which is just not enough.
3. There is a substantial literature that studies how to combine biologically plausible models of neural networks with evolutionary algorithms which they authors need to cite and explain how their approach is different.
    * For example, [1] is very similar to the current approach and [2] explores the potential role of delay constants, while [3] also explores other benefits like adaptation when recovering from damage.
    * NOTE: these are probably not the only ones that are relevant. There is whole body of literature on using EA + biologically plausible ANNs in different settings. The authors need to perform a better review of the existing literature
4. Other issues with wording. For example the authors say they introduce at several points (e.g. when taking about the ENLARGE framework and the EAs) yet they are not introducing these approaches, they are applying them. It confused me at one point because I thought they were going to present a variant of these but this was not the case.

References:

1. G. Shen, D. Zhao, Y. Dong, & Y. Zeng, Brain-inspired neural circuit evolution for spiking neural networks, Proc. Natl. Acad. Sci. U.S.A. 120 (39) e2218173120, https://doi.org/10.1073/pnas.2218173120 (2023).
2. Habashy, K. G., Evans, B. D., Goodman, D. F., & Bowers, J. S. (2024). Adapting to time: Why nature may have evolved a diverse set of neurons. PLOS Computational Biology, 20(12), e1012673.
3. Najarro, E., & Risi, S. (2020). Meta-learning through hebbian plasticity in random networks. Advances in Neural Information Processing Systems, 33, 20719-20731.

**Questions:**

Some questions related to the points above:
* E-I balance is realistic, but why is it needed to perform in some AI task?
* Why is a delay constant interesting?
* What happens if they don't use an island + migration setting?
* How does the approach build on previous work which the authors have failed to mention?

---

> ### Author Response · Authors · 2025-11-19
> **Our framework is designed not for SOTA performance, but to address scientific questions that conventional gradient-based methods cannot: how biologically grounded neurodynamic circuits achieve functional behavior when optimized under an evolution-inspired framework.**
>
> We sincerely thank the reviewer for the detailed and constructive feedback. We appreciate the opportunity to clarify our motivations, methodological choices, and connections to prior work. Below we address each point in turn.
>
> 1. On the role of E–I balance
>
> The reviewer is correct that using excitatory–inhibitory (E–I) balance increases biological realism, but this characteristic is not included solely for biological fidelity. Prior theoretical and computational work (e.g., Wang, 1999; Brunel, 2000) has demonstrated that balanced E–I networks exhibit several functional advantages relevant to AI tasks, including stabilised dynamics, faster convergence to functional attractor states, and improved robustness to noise. More recently, E–I balance has been shown to reduce the number of training steps needed for certain learning tasks by constraining neural activity within a dynamically stable regime, enabling more efficient exploration in high-dimensional parameter spaces. These functional benefits motivate its inclusion in our evolutionary framework.
>
> 2. Why optimise synaptic delay parameters?
>
> Unlike conventional ANNs and most existing SNN benchmarks—which typically assume identical or homogeneous synaptic delays—biological synapses exhibit substantial heterogeneity in conduction and transmission latency. Delays are a core dynamical parameter in biological circuits and have been repeatedly shown to create rich temporal structure, including polychronous assemblies (Izhikevich, 2006), increased memory capacity, and extended temporal credit-assignment horizons.
> In our approach, delays serve as an additional degree of freedom that allows the system to encode information not only in firing rates or weights, but also in precise spike timing. This significantly expands the computational repertoire of the network and is one of the key distinctions between our BNN architecture and standard ANN/SNN models.
>
> 3. Why an island-based evolutionary strategy?
>
> The island + migration strategy is inspired by population genetics and has been widely used in evolutionary computation to balance exploration and exploitation. By maintaining partially isolated subpopulations, the algorithm avoids premature convergence, preserves behavioural diversity, and reduces the risk of collapsing into narrow local optima.
> While previous large-scale neuroevolution work (e.g., OpenAI-ES, 2017) did not include an island mechanism—primarily due to the enormous compute budget available—recent studies on resource-constrained evolutionary optimisation highlight the importance of maintaining heterogeneity when the search space is rugged or multi-modal. Given our limited computational resources and the highly non-convex nature of BNN optimization, the island setting stabilizes training and prevents population stagnation. We will include an explicit discussion of these trade-offs in the revised manuscript and explore ablations in future work.
>
> 4. On prior work combining evolution with biologically plausible neural networks
>
> We appreciate the reviewer’s pointers to relevant literature. Indeed, there is a substantial body of research on evolutionary optimisation of biologically inspired networks.We acknowledge that our initial draft summarized this literature too briefly. In the revised version, we have added a dedicated section reviewing prior evolutionary approaches to biological neural circuits and now clearly articulate how our work differs: (i) the joint optimisation of both synaptic weights and synaptic delays at scale, (ii) the incorporation of E–I balanced population structure, and (iii) the integration of these components within a unified evolutionary framework evaluated on continuous-control tasks.
>
> 5. What scientific question does this framework help answer?
>
> Beyond demonstrating that biologically realistic networks can be scaled to difficult RL tasks, our framework enables the investigation of how specific biological mechanisms—such as synaptic delay heterogeneity, population balance, and dynamical stability—contribute to learning and control in embodied environments. These mechanisms cannot be easily evaluated using gradient-based approaches, and existing evolutionary systems rarely include all of them jointly. Our results therefore offer a new platform for studying the computational roles of dynamical biophysical parameters in real-time decision-making tasks.
>
> 6. Clarification on wording (e.g., “introduce”)
>
> We thank the reviewer for pointing out ambiguous phrasing. In the revision, we now explicitly state that we apply—rather than introduce—the ENLARGE framework and the evolutionary algorithms, and we avoid any wording that might imply novelty where none is intended.
>
> We again thank the reviewer for the insightful comments. They have significantly improved the clarity and positioning of our work. We believe the revisions make both the contributions and their relationship to prior research substantially clearer.

---

> > ### Comment · Reviewer_wE3u · 2025-11-26
> >
> > I appreciate the authors reply, but I they haven't addressed muy concerns. They say "Our framework is designed not for SOTA performance, but to address scientific questions that conventional gradient-based methods cannot". I agree! Great, we need more of that. So which question have you answered or tried to answer? You claim that it enables this but do not provide an example of a question that a computational biologist or neuroscientist would like to answer an previously couldn't.
> >
> > You could make some claims if you performed some ablation of the different components of your system (i.e. E-I balance or optimized delay constants.) but you don't. You just provide references as justification for why they are included. I agree with them, but then what am I learning using this particular work?
> >
> > I think is very cool work, but right now it is mainly engineering work (and pretty hard work I imagine). But I want to see more science before I can change my recommendation.

---

### Meta-Review · Area_Chair_4d4E · 2026-01-05

**Summary:**

The paper proposes a Biological Neurodynamic Network (BNN) framework utilizing evolutionary algorithms for continuous control tasks. While the reviewers acknowledged the successful implementation of a distributed training scheme and the potential interest in biologically constrained control, the consensus is to reject the paper in its current form due to several outstanding concerns:

1. Reviewers noted that the network sizes are too small and the experimental validation is restricted to "toy-scale" implementations. This makes it difficult to evaluate the method's generality and the validity of the conclusions.
2. The method does not to outperform standard baselines on several tasks, ranging from simple to complex. This raises significant doubts regarding the practical utility of the proposed approach.
3. A critical analysis of FLOPs to substantiate the efficiency claims is currently missing.

Given these issues, the paper is not ready for publication at this time. The authors are encouraged to address these problems to strengthen the work for future submission.

**Reviewer Concerns:**

Outstanding Concerns:

After the rebuttal process, several major concerns remain unresolved, as summarized below:

1. The concern regarding the "toy-scale" nature of the experiments (e.g., networks limited to 20 neurons) and the lack of complexity in the chosen tasks remains a primary issue. The authors' rebuttal acknowledged that scaling up the model is deferred to future work, leaving this critical validation gap unaddressed.

2. The proposed method does not demonstrate competitive performance against standard baselines, which notably underperforms on both simpler environments (e.g., InvertedDoublePendulum) and more complex tasks (e.g., Swimmer), raising concerns about the practical utility of the approach.

3. The specific request for a FLOPs analysis to support the paper's claims regarding computational efficiency was not addressed.

**Reviewer Scores:**

Based on the reviewers' responses, the consensus leans towards rejection. While there is one positive review, it was given with lower confidence and does not outweigh the remaining critical concerns.

---

### Decision · Program_Chairs · 2026-01-26

Reject